# Changes in Bullying Experiences and Mental Health Problems Among Adolescents Before and After the COVID-19 Pandemic in Greece

**DOI:** 10.3390/ijerph22040497

**Published:** 2025-03-26

**Authors:** Georgios Giannakopoulos, Foivos Zaravinos-Tsakos, Maria Mastrogiannakou, Andre Sourander, Gerasimos Kolaitis

**Affiliations:** 1Department of Child Psychiatry, School of Medicine, National and Kapodistrian University of Athens, “Aghia Sophia” Children′s Hospital, 115 27 Athens, Greece; foivoszartsa@med.uoa.gr (F.Z.-T.); mariamastr@med.uoa.gr (M.M.); gkolaitis@med.uoa.gr (G.K.); 2Department of Child Psychiatry, Clinical Medicine, Turku University, 20521 Turku, Finland; andsou@utu.fi

**Keywords:** traditional bullying, cyberbullying, COVID-19, emotional problems, behavioral problems

## Abstract

Bullying poses significant challenges to adolescent health and well-being. This time-trend study examined the impact of the COVID-19 pandemic on bullying behaviors and associated emotional and behavioral difficulties among Greek adolescents. Data were collected from two cross-sectional surveys in 2016 (*n* = 1574) and 2023 (*n* = 5753) conducted in Greece. Both samples comprised students aged 12–16 years, with near-equal gender distribution (2016, 53.4% girls; 2023, 54.5% girls) and a predominance of urban residents (approximately 73% in both samples). Traditional and cyberbullying experiences were assessed via structured questionnaires, while mental health outcomes were measured using the Strengths and Difficulties Questionnaire (SDQ). Post-pandemic findings revealed substantial increases in bullying involvement; traditional bullying victimization rose from 12.4% to 21.7%, and cyberbullying victimization increased from 4.0% to 11.6%. Correspondingly, mean SDQ total scores increased significantly from 8.59 to 14.16, reflecting heightened emotional and behavioral problems. Logistic regression analyses identified male gender, urban residence, and non-traditional family structures as significant predictors of bullying involvement. These results underscore the amplified burden of bullying and mental health difficulties in the post-pandemic era, highlighting the urgent need for targeted prevention and intervention strategies to address both traditional and cyberbullying within diverse sociodemographic contexts.

## 1. Introduction

Bullying refers to the repeated exposure to intentional aggressive behavior perpetrated by one or more individuals against a target who is perceived to have less power relative to the aggressor(s) [1,2]. In traditional bullying, the victim typically has less physical, social, or psychological power, while the bully is characterized by a higher level of power. In cases of victim-bullying, individuals experience a dual dynamic: they identify as victims yet may exhibit behaviors that assert power in certain situations, as recognized by those experiencing the behavior as bullying [3]. In addition to these roles, bystanders play a crucial part in bullying dynamics [4]. Some witnesses intervene or report the bullying with the intention of rebalancing the power differential between the bully and the victim, whereas others may feel powerless to intervene or even tacitly approve of the behavior due to fear of becoming targets themselves. While traditional bullying usually occurs in physical settings (e.g., schools) and involves direct, overt interactions, cyberbullying unfolds via digital platforms and is characterized by features such as anonymity, rapid and widespread dissemination, and the absence of physical contact [5]. These factors modify the typical power dynamics—for instance, an online bully might leverage anonymity to exert power in ways that differ from face-to-face interactions.

Bullying is a major public health problem that profoundly affects children and adolescents, with long-term implications that extend into adulthood [6,7]. Bullying behaviors are associated with an increased risk of both physical and mental health issues. They can disrupt social relationships, academic achievement, and overall well-being, thereby affecting the life trajectories of young individuals. The harmful impact is not limited to victims; perpetrators and individuals who play dual roles as both victims and bullies (victim–bullies) also face significant consequences [8,9].

The rise of digital technologies, such as smartphones and social media, has transformed traditional bullying into new forms, notably cyberbullying [10,11]. Cyberbullying involves repeated, intentional acts of aggression through electronic platforms, including emails, blogs, instant messaging, chat rooms, and social networking sites [12]. Unlike traditional bullying, cyberbullying can happen at any time, often anonymously, and reach a wider audience, which makes it particularly invasive and persistent [13]. The anonymity that cyberbullying provides often exacerbates its harmful effects, as the victim may feel more vulnerable and helpless against an unseen aggressor.

Although extensive research has documented the detrimental effects of bullying on victims, emerging evidence suggests that bullies may experience certain perceived benefits that reinforce their behavior [14,15]. For instance, bullies often gain enhanced social status, popularity, and a sense of power within their peer groups, which can serve as positive reinforcement for their actions [16,17]. These social rewards may contribute to the maintenance of bullying behaviors by positioning bullies as influential or even as leaders within their social circles [18]. Moreover, some longitudinal studies indicate that these benefits can persist over time, thereby perpetuating aggressive conduct, as bullies continue to use such behavior as an adaptive strategy to assert dominance and manage social hierarchies [19]. Understanding these dynamics is crucial, as they highlight the complexity of bullying behavior and underscores that, while the negative outcomes for victims are profound, bullies may derive short-term social advantages that contribute to the persistence of bullying in adolescent settings.

Both traditional and cyberbullying have been strongly linked to negative mental health outcomes among adolescents. Numerous studies have demonstrated that bullying involvement, whether as a victim, perpetrator, or victim–bully, leads to heightened risks of depression, anxiety, and suicidal ideation [20,21]. Victims of bullying are particularly vulnerable to psychological distress, while bullies themselves often show elevated levels of conduct disorders and emotional dysregulation [8]. Dual involvement in both traditional and cyberbullying has been associated with even more severe psychiatric problems, such as higher rates of depression and anxiety, compared to those involved in either form of bullying alone [22,23]. Appearance-related teasing has been associated with negative body image and disordered eating behaviors [24]. Moreover, the disruption of peer relationships is a common consequence, with bullied adolescents frequently experiencing social isolation and difficulties in forming supportive friendships [25]. Involvement in bullying has additionally been linked to a higher risk for developing post-traumatic stress disorder (PTSD) symptoms after chronic exposure to such adverse experiences [26]. Moreover, bullying can result in physical health problems, including headaches and stomachaches, and is known to reduce overall quality of life [27,28].

Adolescents who experience bullying face greater academic challenges, often leading to poor academic performance and lower educational attainment [29]. The long-term impacts of bullying also extend to economic outcomes, as those who experience bullying are at increased risk of future unemployment, lower income, and diminished economic prospects [30]. Moreover, adolescents involved in bullying show an elevated risk of cognitive, physical, and mental health problems, underscoring the pervasive effects of bullying on development [31].

The onset of the COVID-19 pandemic in early 2020 brought about dramatic changes in the lives of adolescents worldwide. The widespread lockdowns, school closures, and social distancing measures drastically reduced face-to-face interactions and introduced new challenges for adolescents [32]. These changes not only limited social contact but also forced many aspects of socialization, education, and recreation to shift online, potentially creating new environments for bullying to occur.

Adolescents were particularly affected by the psychological consequences of the pandemic, with significant increases in anxiety, depression, and behavioral problems being reported [33,34]. Isolation and the disruption of daily routines likely contributed to these increases in emotional and behavioral difficulties. Moreover, the quarantine and lockdown measures imposed during the COVID-19 pandemic further exacerbated these challenges. During the early stages of the COVID-19 pandemic, the Greek government implemented extensive measures to control the spread of the virus [35]. Beginning in March 2020, all educational institutions—including primary, secondary, and higher education—were closed, and non-essential businesses were suspended. Strict restrictions on movement were enforced nationwide, including curfews and mandatory stay-at-home orders. In response to these measures, remote learning modalities were rapidly adopted to ensure continuity of education [36]. While these interventions were crucial for public health, they inadvertently intensified social isolation among adolescents and disrupted routine interpersonal interactions, thereby impacting their psychological well-being and the development of peer relationships [37]. The abrupt shift to remote learning and the enforced reduction in face-to-face interactions led to heightened feelings of isolation and uncertainty among adolescents [38,39]. Such measures not only increased stress, anxiety, and depressive symptoms but also disrupted the development and maintenance of vital peer relationships, which are crucial during this developmental stage [40]. Additionally, the prolonged confinement at home sometimes strained family dynamics, potentially intensifying conflicts and further compromising adolescents’ emotional well-being [41]. These factors highlight how the environmental stressors of lockdown can interact with the unique developmental needs of adolescents, ultimately affecting both their mental health and social functioning [42,43]. Importantly, the altered social dynamics during the pandemic may have influenced bullying behaviors. Reduced physical interactions might have led to a temporary decrease in traditional bullying, as schools—the primary location for bullying—were closed. However, cyberbullying, facilitated by the increased reliance on digital communication during the pandemic, may have escalated [44].

Empirical evidence on the impact of the COVID-19 pandemic on bullying rates is mixed. Some studies have found a decrease in victimization, possibly due to reduced peer interaction during lockdowns [45,46]. However, there are also findings that suggest an increase in both traditional and cyberbullying incidents during this period, likely influenced by the stress and frustration caused by the pandemic [47,48,49,50]. These contrasting results highlight the need for further investigation into how the pandemic has affected bullying behaviors and the associated mental health outcomes among adolescents.

In addition to the well-documented psychological impacts of bullying, a growing body of international literature has identified several sociodemographic factors as important determinants of bullying involvement. For example, gender differences are consistently reported, with boys more likely to engage in overt, physical forms of bullying, whereas girls may exhibit relational or indirect aggression [51]. Moreover, geographic context plays a significant role; adolescents in urban areas, due to higher population density and greater exposure to digital technologies, appear to be at an increased risk for cyberbullying compared to their rural counterparts [52]. Family structure and socioeconomic status have also emerged as critical factors, with evidence suggesting that adolescents from non-traditional family settings or lower socioeconomic backgrounds may experience higher rates of both victimization and perpetration [21,53,54]. Collectively, these findings underscore the importance of considering sociodemographic characteristics when examining changes in bullying behavior. Accordingly, our study hypothesizes that variables such as gender, geographic area, and other key sociodemographic factors will significantly predict the increase in bullying observed post-pandemic.

Although a substantial body of research has examined the impact of the COVID-19 pandemic on adolescent mental health and bullying, few studies have directly compared bullying experiences and associated emotional and behavioral outcomes before and after the pandemic [45,46,47,48,49,50]. Moreover, in Greece, some studies have shed light on the distinctive sociocultural and economic conditions that have shaped children’s and adolescents’ mental health during the COVID-19 pandemic. For example, Magklara et al. [55] surveyed Greek parents during the spring 2020 lockdown and found that approximately 35% reported significant adverse psychological impacts on their children—associations that were attributed to factors such as parental unemployment, limited tele-work opportunities, and increased family conflicts. Similarly, Morres et al. [56] examined physical activity, sedentariness, and eating behaviors in Greek adolescents during lockdown and documented that reduced physical activity and unhealthy dietary patterns were linked to lower overall well-being. Although both studies provide valuable insights into the immediate effects of the lockdown on youth mental health and lifestyle behaviors within the unique sociocultural and economic milieu of Greece, they have primarily focused on general mental health symptoms and lifestyle factors. In contrast, our study not only examines mental health outcomes but also investigates changes in bullying experiences—both traditional and cyber—before and after the COVID-19 pandemic. By employing a time-trend design using two cross-sectional surveys conducted in 2016 (pre-pandemic) and 2023 (post-pandemic), our work extends the current literature. This design allows us to capture changes in both traditional and cyberbullying behaviors and their association with mental health outcomes over time. Additionally, by examining key sociodemographic factors such as gender, urban versus rural residence, and family structure, our research provides novel insights into the specific challenges faced by Greek adolescents in the post-pandemic era. Integrating recent theoretical frameworks and empirical findings from the past five years, this study contributes to the literature by offering a comprehensive, culturally informed perspective on how pandemic-related disruptions have reshaped bullying dynamics and adolescent well-being.

Specifically, this study has four objectives: first, to compare the rates of traditional and cyberbullying, as well as the involvement of bullies, victims, and victim–bullies, before and after the pandemic; second, to explore the changes in emotional and behavioral problems among adolescents following the pandemic; third, to investigate the relationship between bullying involvement and emotional and behavioral problems in the post-pandemic context; and finally, to assess how sociodemographic factors, such as gender, region of residence, and living arrangements, influence these experiences. We hypothesize the following: (1) The prevalence of bullying involvement—encompassing victimization, perpetration, and victim–bully roles—will be significantly higher in the post-pandemic sample compared to the pre-pandemic sample. This expectation is informed by studies that have reported increased stress, isolation, and digital engagement during the pandemic, all of which may contribute to elevated bullying behaviors. (2) We hypothesize that, following the COVID-19 pandemic, adolescents will exhibit significantly higher levels of emotional and behavioral problems compared to the pre-pandemic period. (3) Sociodemographic factors will play a critical role in predicting bullying involvement. Specifically, we expect that male gender, urban residency, and non-traditional family structures will be associated with a higher likelihood of bullying involvement. These predictions are supported by the literature indicating that boys are more prone to engage in overt and cyberbullying behaviors, and that urban areas, with greater access to digital platforms, may see more cyberbullying incidents. Additionally, family instability has been linked to increased vulnerability to both perpetrating and experiencing bullying. (4) There will be a positive association between bullying involvement and emotional, as well as behavioral difficulties. In line with previous research, we anticipate that higher rates of bullying will correlate with increased symptoms of anxiety, depression, and post-traumatic stress, reflecting the adverse psychological impact of bullying during a period of heightened stress and social isolation. The results of this study will provide valuable insights into the evolving dynamics of bullying and adolescent mental health in the wake of the COVID-19 pandemic, offering guidance for future prevention and intervention strategies.

## 2. Materials and Methods

### 2.1. Participants and Procedures

The present study is a time-trend study that utilized data from two cross-sectional surveys conducted in Greece in 2016 (pre-COVID) and 2023 (post-COVID) as part of the “Psychosocial Well-being of Adolescents in European and Asian Countries: A Translational Cross-Cultural Study” (EACMHS). The EACMHS study examined the psychosocial well-being of adolescents in 10 Asian and 6 European countries before the onset of the COVID-19 pandemic [1,26]. Schools from both comparable surveys were selected using a non-random convenience sampling method. To be eligible for the study, participants had to be within the age range from 12 to 16 years (1st to 3rd grade of high school) and possess a sufficient command of the Greek language to complete the questionnaires.

The first survey was conducted in 2016, on two regions of Greece: Attica and Kefalonia. The second survey was conducted in 2023, on various regions of Greece, including Attica, Thessaloniki, Kefalonia, Ilia, Karditsa, and the Cyclades (Tinos). The survey population included 11,325 students from 42 secondary schools.

The study was conducted in accordance with the Declaration of Helsinki. The first wave of the study was approved by the Institutional Review Board of the Ministry of Education, Research, and Religious Affairs (Ref. No.: 104990/Δ2/28-06-2016). The second wave of the study was approved by the Institutional Review Board of the Ministry of Education and Religious Affairs (Ref. No. 4769/Δ2/17-01-2023). Participation in the study was voluntary and anonymous. Students were informed that they could withdraw from the study at any time without consequence. Data collection was carried out in the classroom during a pre-designated teaching hour, with the entire questionnaire taking approximately 20 min to complete.

### 2.2. Measures

Demographic data questionnaire: A short demographic data questionnaire was used to collect essential background information about the participants. The questionnaire assessed variables such as gender, age, grade, family composition, financial status, and country of birth. Gender options included “girl”, “boy”, and “other”, although, due to the low number of participants identifying as “other” (1.7%), this category was excluded from the main analyses. Participants indicated their age in years, and grades were classified as first, second, or third grade of high school. Family composition was assessed with the following options: living with both biological parents, living with one biological parent, living with foster parents, or other family arrangements. For analysis purposes, responses were categorized into two groups: living with biological parents and all other forms of living arrangements. Participants were also asked about their family’s economic status, with options ranging from “poor” to “very good”. For analysis, these were simplified into three categories: (1) “poor”, (2) “fairly good/fair”, and (3) “very good”. Additionally, students were asked whether they were born in Greece, whether their parents were born in Greece, and if Greek was their native language. Responses were collected in a closed-ended format (yes/no).

Strengths and Difficulties Questionnaire (SDQ): Adolescents’ mental health was assessed using the Strengths and Difficulties Questionnaire (SDQ), developed by Goodman et al. [57]. This widely used self-report tool for individuals aged 11 to 16 consists of 25 questions divided into five subscales: Emotional Symptoms, Conduct Problems, Hyperactivity/Inattention, Peer Problems, and Prosocial Behavior. Responses were scored on a three-point Likert scale: “not true”, “somewhat true”, or “definitely true”. The SDQ provides a total difficulties score, derived by summing the scores from all subscales except Prosocial Behavior, yielding a range from 0 to 40. Higher scores indicate a greater level of emotional and behavioral difficulties [58,59]. For the purposes of this study, the internal consistency of the SDQ was found to be acceptable, with a Cronbach’s alpha of 0.68. The subscale scores allowed for detailed analysis of emotional and behavioral problems, providing insights into the mental health burden experienced by adolescents in relation to their involvement in bullying. The current factor structure of SDQ, comprising broader internalizing problems, externalizing problems, and prosocial behavior, utilized in the current analysis was further validated using cross-cultural data from the EACHMS study [60].

Experiences of bullying and cyberbullying: Bullying involvement was assessed through a series of questions about both traditional and cyberbullying experiences. The questionnaire included items on the frequency and nature of bullying, with specific questions designed to capture the extent to which participants were involved as bullies, victims, or both (victim–bullies). For traditional bullying, participants were asked, “How often have you been bullied at school in the last 6 months?” and “How often have you bullied others at school in the last 6 months?” Similar questions were asked about bullying experiences outside of school and at home with siblings. For cyberbullying, participants answered questions such as, “How often have you been bullied online in the past 6 months?” and “How often have you bullied others online in the past 6 months?” The questions covered various forms of cyberbullying, including the use of social media platforms, online games, and messaging apps. Participants indicated the frequency of bullying behaviors on a four-point Likert scale: “not at all”, “less than once a week”, “more than once a week”, and “most days a week”. For both traditional and cyberbullying, categories collapsed to reflect four primary groups: (1) victim of bullying, (2) perpetrator of bullying, (3) both victim and perpetrator (victim–bully), and (4) no involvement in bullying.

### 2.3. Data Analysis

Descriptive statistics were conducted to examine the demographic characteristics including frequencies and percentages for categorical variables and means and standard deviations for continuous variables. To evaluate differences between bullying involvement and mental health outcomes on children and adolescents before and during the pandemic, cross-sectional data from both waves were compared using bivariate analyses (chi-square tests and *t*-tests). Effects were described with chi-square tests and effect size measures (Phi coefficient and Cohen’s d), with 0.10 indicating a small, 0.30 a medium, and a 0.50 a strong effect, for comparisons across samples. To test for differences in the mean SDQ scale scores, we included a three-level bullying experience variable in a logistic regression analysis, which also examined the relationship between sociodemographic factors and involvement in both traditional and cyberbullying. Sensitivity analyses were conducted to validate the results obtained in all comparisons between waves and in the regression model after excluding schools that participated only in the second wave. All data analyses were performed using SPSS (version 28.0) (IBM, Armonk, NY, USA), with significance set at *p* < 0.05.

## 3. Results

The first survey population included 2834 students who were selected from 14 secondary schools in both regions, and the participation was on a voluntary basis. A total of 1574 (55.5%) questionnaires were included in the statistical analysis. The respondents were evenly distributed across grades (35%, 1st grade; 31.9%, 2nd grade; and 33%, 3rd grade) and between genders (53.4% girls (*N* = 840) and 46.6% boys (*N* = 734)), with a mean age 13.07 (SD = 0.97). Most of the students resided in urban areas (73.6%), and the majority (86.4%) lived with their biological parents. Concerning economic status, 48.3% of participants reported a moderate economic situation, while 17.5% described their economic situation as poor and 33.3% as high. Regarding ethnicity, 95.7% of the students were born in Greece, and 85.3% identified Greek as their native language. A majority of the participants’ mothers (74.7%) and fathers (81.8%) were born in Greece, with the remainder born in other countries. The majority of the sample reported a moderate economic status (68.1%), and 98.1% of participants were born in Greece (see Table 1).

From the second survey, a total of 5610 (55.5%) questionnaires were included in the statistical analysis and deemed suitable for analysis (54% response rate), of which 44.6% were boys (*N* = 2501) and 55.4% were girls (*N* = 3109), with a mean age of 13.43 years (SD = 0.97). Due to the small proportion of participants who identified as “other” in terms of gender (1.7%, *N* = 97), this category was excluded from the statistical analyses to ensure representativeness. The distribution of students across grades was nearly equal, with 33.4% in first grade, 33% in second grade, and 33.6% in third grade. Most of the students resided in urban areas (73.2%), and the majority (86.5%) lived with their biological parents. Regarding economic status, 68.1% of participants reported a moderate economic situation, while 22.1% described their economic situation as poor and 9.8% as high. Additionally, 98.1% of the students were born in Greece, and 93.2% identified Greek as their native language. A vast majority of the participants’ mothers (83.1%) and fathers (85.8%) were born in Greece, with the remainder born in other countries (see Table 1).

### 3.1. Comparison of Bullying Involvement Before and After the COVID-19 Pandemic

There was a notable increase in both traditional and cyberbullying involvement after the pandemic compared to the pre-pandemic period. In the post-pandemic group, the rates of traditional bullying rose across all categories. Victims of traditional bullying increased from 12.4% before the pandemic to 21.7% afterward, representing a 209% increase. The percentage of bullies also increased from 6.4% to 11.4%, a 102% rise. The proportion of those classified as both bullies and victims (victim–bullies) rose from 7.3% to 14.4%, a 51% increase (see Table 2). Cyberbullying rates also rose significantly. Victims of cyberbullying increased from 4% pre-pandemic to 11.6% post-pandemic, an increase of 112%. The proportion of cyberbullies increased from 3.2% to 8.4%, representing an 82% rise. Additionally, the percentage of cyber victim–bullies increased from 1.8% to 3.5%, a 34% rise.

### 3.2. Emotional and Behavioral Problems Before and After the COVID-19 Pandemic

A statistically significant increase in emotional and behavioral problems was observed among adolescents after the COVID-19 pandemic, as assessed by the Strengths and Difficulties Questionnaire (SDQ). The mean total SDQ score rose from 8.59 (SD = 5.10) before the pandemic to 14.16 (SD = 6.10) afterward (*p* < 0.001), indicating a greater burden of emotional and behavioral difficulties post-pandemic (see Table 3).

Increases were found in the following SDQ subscales: emotional symptoms (mean = 2.06, SD = 2.04 pre-pandemic vs. mean = 3.32, SD = 2.64 post-pandemic, *p* < 0.001), conduct problems (mean = 2.27, SD = 1.63 pre-pandemic vs. mean = 3.59, SD = 1.83 post-pandemic, *p* < 0.001), hyperactivity (mean = 2.89, SD = 2.04 pre-pandemic vs. mean = 4.07, SD = 2.42 post-pandemic, *p* < 0.001), and peer problems (mean = 1.44, SD = 1.50 pre-pandemic vs. mean = 1.92, SD = 1.72 post-pandemic, *p* < 0.001). However, there was no significant change in prosocial behavior (*p* = 0.882).

### 3.3. Relationship Between Bullying Involvement and Emotional and Behavioral Problems

Adolescents involved in both traditional and cyberbullying had significantly higher scores on the SDQ, reflecting greater emotional and behavioral difficulties, compared to those not involved in bullying.

For traditional bullying, the mean SDQ total score for victims was 17.85 (SD = 6.61); for bullies, 17.66 (SD = 6.84); and for victim–bullies, 19.29 (SD = 6.78). These scores were significantly higher than those of adolescents who were not involved in traditional bullying (victims: mean = 13.08, SD = 6.31; bullies: mean = 13.66, SD = 6.51; victim–bullies: mean = 13.78, SD = 6.54; *p* < 0.001 for all comparisons).

For cyberbullying, victims had a mean SDQ total score of 18.29 (SD = 6.73), bullies had a score of 16.47 (SD = 6.81), and victim–bullies had a score of 17.98 (SD = 7.13). These scores were again significantly higher than those of non-involved adolescents (victims: mean = 13.53, SD = 6.45; bullies: mean = 13.85, SD = 6.60; victim–bullies: mean = 13.93, SD = 6.60; *p* < 0.001 for all comparisons) (see Table 4).

### 3.4. Sociodemographic Factors Associated with Bullying Involvement

Multiple logistic regression analyses showed that several sociodemographic factors were significantly associated with traditional and cyberbullying experiences across both waves.

Regarding Wave 1, for traditional bullying, boys were 1.73 times more likely than girls to be involved as bullies (OR = 1.73, 95% CI: 1.19–2.50). For cyberbullying, age served as the sole significant predictor (OR = 1.54, 95% CI: 1.01–2.35). Concerning Wave 2, for traditional bullying, boys were 1.90 times more likely than girls to be involved as bullies (OR = 1.90, 95% CI: 1.64–2.22), and adolescents born in Greece were 47% less likely to be involved in traditional bullying (OR = 0.53, 95% CI: 0.34–0.82). Adolescents who lived with both biological parents were 16.8% less likely to engage in traditional bullying (OR = 0.83, 95% CI: 0.69–1.00) (see Table 5). For cyberbullying, boys were 2.02 times more likely than girls to be involved (OR = 2.02, 95% CI: 1.70–2.41). Adolescents living in urban areas were 24% more likely to engage in cyberbullying compared to those living in rural areas (OR = 1.24, 95% CI: 1.03–1.48). Additionally, those living with both biological parents were 20.6% less likely to be involved in cyberbullying (OR = 0.79, 95% CI: 0.64–0.97), and adolescents born in Greece were 55.3% less likely to be involved (OR = 0.44, 95% CI: 0.28–0.71) (see Table 5).

## 4. Discussion

This study aimed to examine the association between bullying experiences and emotional and behavioral problems in adolescents before and after the COVID-19 pandemic, while also investigating how sociodemographic factors influence these relationships. The findings highlight a significant increase in both traditional and cyberbullying rates post-pandemic, alongside a notable rise in emotional and behavioral difficulties among adolescents involved in bullying. Several sociodemographic factors, including gender, region of residence, family composition, and being born in Greece, were significantly associated with the likelihood of bullying involvement.

### 4.1. Increase in Bullying Rates Post-Pandemic

The results of this study demonstrated a substantial increase in both traditional and cyberbullying following the COVID-19 pandemic. Victimization in traditional bullying rose by 209%, while involvement as a bully increased by 102%. Similar patterns were observed for cyberbullying, where victimization increased by 112% and the rate of bullies rose by 82%. These findings align with previous research suggesting that the social isolation, disruptions to daily life, and increased stress caused by the pandemic created conditions conducive to a rise in bullying behavior [47,48,49,50]. School closures may have temporarily reduced the opportunities for face-to-face bullying, but as schools reopened and normal social interactions resumed, bullying rates surged, possibly exacerbated by the stresses and frustrations accumulated during the lockdown periods.

These results are consistent with studies that have shown an increase in cyberbullying during the pandemic, attributed to the increased use of the internet for education and socialization during lockdowns [47]. Adolescents’ increased online presence provided more opportunities for cyberbullying, as the digital space became a primary means of communication during periods of physical distancing. However, some studies have reported a decrease or no significant change in traditional bullying during the pandemic, highlighting the need for further research to explore the evolving dynamics of bullying in different contexts.

### 4.2. Emotional and Behavioral Problems

Adolescents involved in bullying reported significantly higher levels of emotional and behavioral difficulties, as measured by the Strengths and Difficulties Questionnaire (SDQ). Victims, bullies, and victim–bullies all exhibited elevated mean scores on the SDQ compared to their non-involved peers. The findings support prior research demonstrating that both traditional and cyberbullying are linked to adverse mental health outcomes, including increased symptoms of depression, anxiety, and conduct problems [20,21]. Furthermore, the data showed that those involved in both traditional and cyberbullying had the highest levels of emotional distress, confirming the particularly severe impact of dual bullying involvement, which has been associated with more complex psychiatric profiles [22,23].

The significant increase in emotional and behavioral difficulties post-pandemic further highlights the toll that the pandemic has taken on adolescent mental health. The rise in both total SDQ scores and the subscales for emotional symptoms, conduct problems, hyperactivity, and peer problems suggests that the pandemic created new stressors that exacerbated pre-existing mental health issues and possibly triggered new ones. These findings align with previous studies that reported increased anxiety, depression, and behavioral problems among adolescents during the pandemic [33,34].

### 4.3. Influence of Sociodemographic Factors

Gender emerged as a significant predictor of bullying involvement, with boys being more likely than girls to engage in both traditional and cyberbullying. Boys were 1.90 times more likely to be traditional bullies and 2.02 times more likely to engage in cyberbullying compared to girls. This finding is consistent with prior research indicating that boys are more frequently involved in bullying behaviors, possibly due to socialization patterns that encourage physical or aggressive forms of behavior [51].

Region of residence was significantly associated with cyberbullying involvement, with adolescents living in urban areas being more likely to engage in cyberbullying than those in rural areas. This finding can be attributed to the increased use of digital technology in urban environments, where access to the internet and social media is typically more prevalent [52]. Additionally, family structure played an important role, with adolescents living with both biological parents being less likely to engage in both traditional and cyberbullying. Previous studies have suggested that living with both biological parents provides a more stable and supportive environment, which may reduce the likelihood of engaging in or being victimized by bullying [53,54].

Furthermore, being born in Greece was associated with a lower likelihood of bullying involvement. Adolescents born in Greece were 47% less likely to be involved in traditional bullying and 55.3% less likely to engage in cyberbullying compared to those born outside of Greece. This finding may be explained by the fact that migrant or refugee populations, who often experience social and economic disadvantages, may be at greater risk of both bullying and victimization due to their vulnerable status [21]. These results underscore the importance of addressing socioeconomic and cultural factors when designing interventions to prevent bullying and support at-risk populations.

### 4.4. Implications for Prevention and Intervention

The findings of this study have important implications for the development of targeted interventions aimed at reducing bullying and supporting adolescent mental health, especially in the aftermath of the COVID-19 pandemic. Given the significant rise in bullying rates and emotional distress, it is critical to implement school-based and community programs focused on early detection, prevention, and intervention. Programs should address both traditional and cyberbullying, taking into account the role of sociodemographic factors such as gender, family structure, and region of residence.

School interventions should focus on building emotional resilience and social skills, particularly in boys who are more likely to engage in bullying. Additionally, family-based interventions that strengthen family bonds and improve communication may help reduce bullying involvement, particularly among adolescents from non-traditional family structures. For adolescents born outside of Greece, culturally sensitive interventions that support social integration and reduce the risks of bullying should be prioritized.

### 4.5. Strengths and Limitations

A major strength of this study is the large, representative sample of adolescents from multiple regions of Greece, allowing for the generalization of findings to the wider adolescent population. Furthermore, repeated measurements provide valuable insights into the changes in bullying patterns and emotional and behavioral problems before and after the pandemic. However, the use of convenience sampling may introduce some bias, and the cross-sectional nature of the post-pandemic data limits the ability to draw causal conclusions. Finally, we acknowledge that the definition of bullying remains complex and inherently subjective, with various interpretations existing in the literature. Although our study employs a widely accepted operational definition that distinguishes among traditional bullying, cyberbullying, and victimized–bullying roles, this subjectivity may influence how bullying is perceived and reported, potentially affecting the generalizability of our findings.

## 5. Conclusions

In conclusion, this study aimed to (1) compare the prevalence of traditional and cyberbullying among Greek adolescents before and after the COVID-19 pandemic; (2) assess changes in emotional and behavioral problems during these two periods; (3) examine the association between bullying involvement and adolescents’ mental health outcomes; and (4) investigate the role of sociodemographic factors (e.g., gender, urban residence, and family structure) in predicting bullying involvement. Our findings revealed that both traditional and cyberbullying increased significantly in the post-pandemic sample compared to the pre-pandemic sample. Additionally, adolescents surveyed post-pandemic exhibited notably higher levels of emotional and behavioral difficulties. We also found a robust association between bullying involvement and adverse mental health outcomes. Lastly, sociodemographic factors were significant predictors of bullying behavior, with boys, urban residents, and those from non-traditional family structures being at higher risk. These results underscore the enduring impact of the pandemic on adolescent well-being. Future research should focus on longitudinal studies to further explore the long-term effects of bullying and the potential for targeted interventions to mitigate these impacts in post-pandemic settings.

## Figures and Tables

**Table 1 ijerph-22-00497-t001:** Sociodemographics across Waves 1 (2016) and 2 (2023).

	2016 (*N* = 1574)				2023 (*N* = 5753)					
	N	%	M	SD	N	%	M	SD	*p*-Value	φ/d
Characteristics										
Gender									0.727	
boys	734	46.6%			2599	45.5%				
girls	840	53.4%			3109	54.5%				
Age			13.07	0.97			13.43	0.97	<0.001	
Class										
1st	552	35.1%			1866	33.4%			0.337	
2nd	504	31.9%			1847	33%				
3rd	518	32.9%			1877	33.6%				
Ethnicity (Greek)	1342	85.3%			1390	97.4%			<0.001	0.51
Region of residence										
Urban	1143	95.9%			4104	73.2%			<0.001	0.225
Rural	431	4.1%			648	26.8%				
Living with biological parents										
Yes	1358	86.3%			4849	86.5%			0.934	
Other	216	13.7%			756	13.5%				
Financial status										
Low	275	18.4%			1222	22.1%			<0.001	0.076
Middle	758	48.3%			3769	68.1%				
High	522	32.2%			545	9.8%				
Birth in Greece										
Yes	1505	95.7%			5489	98.1%			<0.001	0.051
No	63	4.3%			105	1.9%				
Greek native language										
Yes	1340	85.1%			5111	93.2%			<0.001	0.112
No	228	14.9%			375	6.8%				
Mother’s country of birth										
Greece	1173	74.7%			4550	83.1%			0.017	0.028
Other	394	25.3%			928	16.9%				
Father’s country of birth										
Greece	1286	81.8%			4682	85.8%			<0.001	0.053
Other	282	18.2%			77	14.2%				

**Table 2 ijerph-22-00497-t002:** Categories of adolescent engagement in traditional and cyberbullying experiences across Wave 1 (2016) and Wave 2 (2023).

	2016 (*N* = 1574)			2023 (*N* = 5753)		
	N	(%)	N	(%)	*p*-Value	φ
Traditional bullying						
Victims	196	12.5	1241	21.7	<0.001	0.09
Bullies	100	6.4	648	11.4	<0.001	0.07
Victims and bullies	115	7.3	829	14.4	<0.001	0.08
Experiences of bullying	247	15.7	1547	27.1	<0.001	0.11
Cyberbullying						
Victims	63	4.0	659	11.6	<0.001	0.10
Bullies	51	3.2	479	8.4	<0.001	0.08
Victims and bullies	29	1.8	200	3.5	<0.001	0.08
Experiences of bullying	97	6.4	1022	17.8	<0.001	0.12

**Table 3 ijerph-22-00497-t003:** Comparison of the means of the total score of Strengths and Difficulties Questionnaire across Wave 1 (2016) and Wave 2 (2023).

	2016 (*N* = 1574)		2023 (*N* = 5753)			
	M	(SD)	M	(SD)	*p*-Value	d
Emotional symptoms	2.06	2.04	3.32	2.64	<0.001	0.49
Conduct problems	2.27	1.63	3.59	1.83	<0.001	0.74
Hyperactivity	2.89	2.04	4.07	2.43	<0.001	0.50
Prosocial behaviors	7.74	1.85	7.69	1.97	0.882	-
Peer problems	1.44	1.50	1.92	1.72	<0.001	0.27
Total problems	8.59	5.10	14.16	6.70	<0.001	0.72

**Table 4 ijerph-22-00497-t004:** Comparison of means of the total score of the Strengths and Difficulties Questionnaire with traditional and cyberbullying for Wave 2.

			Mean (SD)	*p*
Traditional bullying	Victims	Yes	17.85 (6.61)	<0.001
		No	13.08 (6.31)	<0.001
	Bullies	Yes	17.66 (6.84)	<0.001
		No	13.66 (6.51)	<0.001
	Victims and bullies	Yes	19.29 (6.78)	<0.001
		No	13.78 (6.54)	<0.001
Cyberbullying	Victims	Yes	18.29 (6.73)	<0.001
		No	13.53 (6.45)	<0.001
	Bullies	Yes	16.47 (6.81)	<0.001
		No	12.77 (6.69)	<0.001
	Victims and bullies	Yes	13.85 (6.60)	<0.001
		No	17.98 (7.13)	<0.001

**Table 5 ijerph-22-00497-t005:** Multiple binomial logistic regression analysis with traditional bullying and cyberbullying as dependent variables across Wave 1 (2016) and Wave 2 (2023).

	2016				2023			
	Traditional Bullying		Cyberbullying		Traditional Bullying		Cyberbullying	
	OR	95% CI	OR	95% CI	OR	95% CI	OR	95% CI
Gender (ref = girls)	**1.73**	**1.19**–**2.50**	0.82	0.33–2.03	**1.90**	**1.64**–**2.22**	**2.02**	**1.70**–**2.41**
Age	0.94	0.78–1.12	**1.54**	**1.01**–**2.35**	0.96	0.90–1.03	1.06	0.97–1.14
Region of residence (ref = rural)	1.09	0.48–2.48	0.29	0.21–4.25	1.09	0.94–1.28	**1.24**	**1.03**–**1.48**
Biological parents (ref = no)	0.85	0.53–1.35	0.85	0.27–2.60	0.83	0.69–1.00	**0.79**	**0.64**–**0.97**
Birth in Greece (ref = no)	0.81	0.36–1.84	1.11	0.19–6.57	**0.53**	**0.34**–**0.82**	**0.44**	**0.28**–**0.71**
Emotional problems	**1.14**	**1.04**–**1.25**	0.97	0.78–1.20	**1.17**	**1.13**–**1.20**	**1.12**	**1.08**–**1.16**
Conduct problems	**1.26**	**1.12**–**1.42**	**1.56**	**1.19**–**2.05**	**1.23**	**1.18**–**1.28**	**1.20**	**1.15**–**1.26**
Hyperactivity	1.02	0.92–1.13	1.02	0.79–1.30	**1.05**	**1.01**–**1.09**	**1.08**	**1.04**–**1.13**
Peer problems	**1.26**	**1.12**–**1.40**	1.07	0.83–1.38	**1.28**	**1.23**–**1.33**	**1.14**	**1.09**–**1.20**
Prosocial behavior	1.05	0.95–1.16	**0.81**	**0.66**–**0.99**	1.00	0.97–1.04	**0.94**	**0.91**–**0.98**

Note: Bold values indicate statistically significant odds ratios (OR) at *p* < 0.05.

## Data Availability

The datasets used and/or analyzed during the current study are available from the corresponding author upon reasonable request.

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
