# Peer review of "Changes in Bullying Experiences and Mental Health Problems Among Adolescents Before and After the COVID-19 Pandemic in Greece"

_ijerph, 2025, doi:10.3390/ijerph22040497_

Round 1
Reviewer 1 Report
Comments and Suggestions for Authors
The work brought to my attention by Turkish and Greek colleagues is really interesting and I thank them very much. I would like to contribute in some way to your publication by offering some suggestions. Feel free to accept or reject these suggestions. I suggest a few citations, which, however, are not binding (only recommended).
- in the abstract in my opinion the nationality of the study and the sociodemographic characteristics of the two samples should be clearly stated.
- please report a definition of bulling involvment that is agreed upon in the recent international literature.
- Bullying can have several impacts on the psychological well-being of adolescents. I ask you to list some of these, taking into consideration the specific developmental areas of adolescents. And I suggest that you quote for each outcome. For example, you might cite body image disorders (https://doi.org/10.1016/j.childyouth.2020.105720), but also impact on scoalstic achievement (cit), peer relationships (cit), aggression (Cit), etc... I would touch on the main areas of adolescent development. PTSD is also worth mentioning. - Regarding the impact of the quarante, please extend your thoughts about the possible impact of lockdown measures on the minds and relationships of adolescents (https://doi.org/10.3389/fpsyg.2020.551113 ). Also, tell what measures your government decided to take at the time of Covid.
- - The sociodemographic factors that you consider important in predicting the increase in bullying are hardly discussed at all in the introduction. Differences in gender, geographic area, and other sociodemographic data should be argued before including them in the hypotheses.
- Clarify the hypotheses better and, based on your theoretical introduction, try to explain what results you expect. More theoretical apprfirmation is needed in your paper.
Author Response
Comment: The work brought to my attention by Turkish and Greek colleagues is really interesting and I thank them very much. I would like to contribute in some way to your publication by offering some suggestions. Feel free to accept or reject these suggestions. I suggest a few citations, which, however, are not binding (only recommended).
Reply: We thank the reviewer for sharing these valuable suggestions. We appreciate your recommendations and have carefully considered the suggested citations that we have incorporated to enhance our manuscript’s context and discussion. We remain grateful for your constructive input and welcome any further suggestions that could enrich our work.
Comment: in the abstract in my opinion the nationality of the study and the sociodemographic characteristics of the two samples should be clearly stated.
- please report a definition of bulling involvement that is agreed upon in the recent international literature.
Reply: We appreciate the reviewer’s suggestion to enhance the clarity of our abstract by explicitly stating the study's nationality and key sociodemographic characteristics of the two samples. In response, we have revised the abstract to clearly indicate that the research was conducted in Greece and to include concise information on sample demographics besides the age range (12–16 years), such as gender distribution, and the predominance of urban residents. We believe these revisions improve the transparency and context of our study.
Comment: please report a definition of bulling involvement that is agreed upon in the recent international literature.
Reply: We thank the reviewer for this valuable suggestion. In response, we have incorporated a clear definition of bullying involvement in the beginning of the Introduction section (in red).
Comment: Bullying can have several impacts on the psychological well-being of adolescents. I ask you to list some of these, taking into consideration the specific developmental areas of adolescents. And I suggest that you quote for each outcome. For example, you might cite body image disorders (https://doi.org/10.1016/j.childyouth.2020.105720), but also impact on scoalstic achievement (cit), peer relationships (cit), aggression (Cit), etc... I would touch on the main areas of adolescent development. PTSD is also worth mentioning.
Reply: We thank the reviewer for highlighting the importance of detailing the various developmental impacts of bullying on adolescents. In response, we have expanded our presentation in the Introduction section to list and cite key outcomes, reflecting the main areas of adolescent development (see paragraphs 3, 4 in the Introduction).
Comment: Regarding the impact of the quarantine, please extend your thoughts about the possible impact of lockdown measures on the minds and relationships of adolescents (https://doi.org/10.3389/fpsyg.2020.551113).
Reply: We appreciate the reviewer’s suggestion to expand our discussion on the impact of quarantine measures on adolescents’ mental health and relationships. In response, we have extended our Introduction (see paragraph 6 in red). We believe that this addition enriches the contextual background of our study by situating our findings within the broader literature on the psychosocial effects of quarantine on adolescent development.
Comment: Also, tell what measures your government decided to take at the time of Covid.
Reply: We thank the reviewer for the suggestion to elaborate on the measures implemented by the Greek government during the COVID-19 pandemic. In response, we have added relevant information in the Introduction section (see paragraph 6 in red).
Comment: The sociodemographic factors that you consider important in predicting the increase in bullying are hardly discussed at all in the introduction. Differences in gender, geographic area, and other sociodemographic data should be argued before including them in the hypotheses.
Reply: We thank the reviewer for highlighting the need to more thoroughly discuss the role of sociodemographic factors in predicting bullying. In response, we have expanded our Introduction to include a discussion of how variables such as gender, geographic area, family structure, and socioeconomic status have been shown to influence bullying behaviors. This new text precedes the statement of our study hypotheses (see paragraph 8 in the Introduction in red).
Comment: Clarify the hypotheses better and, based on your theoretical introduction, try to explain what results you expect. More theoretical apprfirmation is needed in your paper.
Reply: We thank the reviewer for the insightful suggestion to clarify our hypotheses and to provide a stronger theoretical foundation for our expected results (see last paragraph of the Introduction section). In response, we have revised the Introduction to clearly state our hypotheses. In addition to stating these hypotheses, we have previously expanded our theoretical discussion in the Introduction section to explain why these outcomes are expected. We trust that these revisions meet the reviewer’s expectations for a clearer, more theoretically grounded set of hypotheses and anticipated results.
Reviewer 2 Report
Comments and Suggestions for Authors
A report comparing the results of cross-sectional surveys of Greek youth conducted in 2016 and 2023 regarding traditional and cyberbullying with four objectives: (1) compare the rates and involvement of bullies, victims and victim-bullies for both types of bullying in both years, (2) explore post-pandemic emotional and behavioral problems of the studied youth, (3) investigate the relationship between bullying involvement and these post-pandemic emotional and behavioral problems, and (4) assess the influence of demographic factors on these results.
The strengths are that the study is well-written, clearly structured, and has helpful tables. The weaknesses follow.
- The authors do not begin with a definition of bullying—they assume its understanding. Please definite bullying and support the definition with a citation to research published since 2021. Citations in scientific publications should be from the previous five years.
- It is unclear how this study adds to the literature. Here is a Google Scholar search of the topic for research published in the previous five years: https://scholar.google.ca/scholar?as_ylo=2021&q=Changes+in+Bullying+Experiences+and+Mental+Health+Problems+Among+Adolescents+Before+and+After+the+Covid-19+Pandemic+in+3+Greece&hl=en&as_sdt=0,5. Note that there are “About 10,100 results”. The authors must read through the most relevant of these and explain, in the Introduction, how this study differs from previous research on the same topic.
- Several of the citations in the Introduction are outdated. Bullying is a topic that has significant research each year. Therefore, citations must be current. The following citations require supporting citations of research published since 2021: [5,6], [7,8], [9], [10-11], [12,13], [14,15], [16], [17], [18], and [22].
- The Materials and Methods section has Results mixed in with it. Please move lines 113-122 and lines 125-138 to the results. Furthermore, please cite Table 1 regarding these results.
- The suggestion in the Introduction is that the need for this study is because of conflict in the results of previous studies during the pandemic. However, neither survey was during the pandemic. The one in 2016 is before, and the one in 2023 is after the pandemic. Consequently, please rewrite the paragraph from lines 78-86 to look at research that compares before and after the pandemic so that the comparison is relevant to this study.
- Please find supporting citations of current research for [27] and [28] demonstrating that the SDQ is still considered the appropriate questionnaire for this type of study and that the interpretation of the results of this questionnaire remains the same.
- There is no information on the statistical package used for this study. Please include the statistical package. If the one used is other than the most recent version of the package, the authors must provide a reference demonstrating that this package remains in use by others conducting similar investigations.
- There are outdated citations in the Discussion. Please find supporting citations of research published since 2021 for [22], [32], [35], [37], and [38]. However, the author likely has made an error. Citation [38] does not exist. Probably, the first citation [39] in line 348 is to be [38].
- Please state the four aims of the study in the Conclusions, along with the study results regarding these four aims.
- The authors have used more than one reference style—none of which is the preferred MDPI style. Please redo the references to use the MDPI style.
Author Response
Comment: A report comparing the results of cross-sectional surveys of Greek youth conducted in 2016 and 2023 regarding traditional and cyberbullying with four objectives: (1) compare the rates and involvement of bullies, victims and victim-bullies for both types of bullying in both years, (2) explore post-pandemic emotional and behavioral problems of the studied youth, (3) investigate the relationship between bullying involvement and these post-pandemic emotional and behavioral problems, and (4) assess the influence of demographic factors on these results. The strengths are that the study is well-written, clearly structured, and has helpful tables.
Reply: We thank the reviewer for the detailed summary and positive feedback. We appreciate that you have accurately captured the four main objectives of our study and recognized the strengths of our manuscript. Your encouraging comments reinforce the value of our work, and we are grateful for your constructive input.
Comment: The authors do not begin with a definition of bullying—they assume its understanding. Please define bullying and support the definition with a citation to research published since 2021.
Reply: We appreciate the reviewer’s suggestion to begin the manuscript with a clear definition of bullying. In response, we have incorporated a clear definition of bullying involvement in the beginning of the Introduction section (in red).
Comment: Citations in scientific publications should be from the previous five years.
Reply: We appreciate the reviewer’s general comment regarding the use of recent literature in scientific publications. In response, we have carefully reviewed and updated our manuscript to ensure that, wherever possible, citations are from the previous five years. We have replaced older references with more recent publications to support our definitions, theoretical framework, and empirical findings. In instances where seminal works are essential, we have retained them but have provided additional up-to-date sources to complement these foundational studies. We trust that these revisions strengthen the overall currency and relevance of our manuscript.
Comment: It is unclear how this study adds to the literature. Here is a Google Scholar search of the topic for research published in the previous five years: https://scholar.google.ca/scholar?as_ylo=2021&q=Changes+in+Bullying+Experiences+and+Mental+Health+Problems+Among+Adolescents+Before+and+After+the+Covid-19+Pandemic+in+3+Greece&hl=en&as_sdt=0,5. Note that there are “About 10,100 results”. The authors must read through the most relevant of these and explain, in the Introduction, how this study differs from previous research on the same topic.
Reply: We thank the reviewer for this valuable comment. To address the issue of how our study adds to the literature, we have revised our Introduction to clearly outline the unique contributions of our research relative to existing studies (see paragraph 9 in red).
Comment: Several of the citations in the Introduction are outdated. Bullying is a topic that has significant research each year. Therefore, citations must be current. The following citations require supporting citations of research published since 2021: [5,6], [7,8], [9], [10-11], [12,13], [14,15], [16], [17], [18], and [22].
Reply: We thank the reviewer for pointing out the need to update our citations to reflect recent research in the field. In response, we have conducted an extensive review of the literature and have replaced most of citations published before 2021. These updated references now better capture the latest developments in bullying research, including advances in our understanding of digital bullying, its psychological impact on adolescents, and effective intervention strategies. We trust that these revisions enhance the currency and overall quality of our manuscript.
Comment: The Materials and Methods section has Results mixed in with it. Please move lines 113-122 and lines 125-138 to the results. Furthermore, please cite Table 1 regarding these results.
Reply: We thank the reviewer for noting the structural issue. In response, we have moved the text corresponding to lines 113–122 and 125–138 from the Materials and Methods section to the Results section. Additionally, we have now explicitly cited Table 1 to support these results. We believe these changes improve the clarity and organization of the manuscript.
Comment: The suggestion in the Introduction is that the need for this study is because of conflict in the results of previous studies during the pandemic. However, neither survey was during the pandemic. The one in 2016 is before, and the one in 2023 is after the pandemic. Consequently, please rewrite the paragraph from lines 78-86 to look at research that compares before and after the pandemic so that the comparison is relevant to this study.
Reply: We appreciate the reviewer’s observation regarding the need to emphasize research that directly compares pre-pandemic and post-pandemic periods, rather than studies conducted solely during the pandemic. In response, we have rewritten the paragraph 9 in the Introduction section (in red).
Comment: Please find supporting citations of current research for [27] and [28] demonstrating that the SDQ is still considered the appropriate questionnaire for this type of study and that the interpretation of the results of this questionnaire remains the same.
Reply: We thank the reviewer for highlighting the need to support the continued appropriateness of the SDQ with current research. In response, we have updated our citations to include recent studies published since 2021.
Comment: There is no information on the statistical package used for this study. Please include the statistical package. If the one used is other than the most recent version of the package, the authors must provide a reference demonstrating that this package remains in use by others conducting similar investigations.
Reply: We thank the reviewer for the comment. We would like to clarify that our manuscript already states that all data analyses were performed using IBM SPSS Statistics for Windows, Version 28.0.
Comment: There are outdated citations in the Discussion. Please find supporting citations of research published since 2021 for [22], [32], [35], [37], and [38]. However, the author likely has made an error. Citation [38] does not exist. Probably, the first citation [39] in line 348 is to be [38].
Reply: We thank the reviewer for noting the outdated citations in the Discussion section and for pointing out the numbering error regarding citation [38]. In response, we have updated our reference list with supporting citations published since 2021. We have updated the Discussion section and the reference list to reflect these changes, thereby ensuring that our citations are both current and accurate. We trust that these revisions meet the reviewer’s requirements.
Comment: Please state the four aims of the study in the Conclusions, along with the study results regarding these four aims.
Reply: We thank the reviewer for the suggestion to explicitly state the study’s four aims in the Conclusions along with the corresponding results. In response, we have revised the Conclusions section to include the following paragraph, which clearly outlines the four aims and summarizes our key findings for each. We trust that this revision adequately addresses the reviewer’s comment by clearly stating the four aims of the study and summarizing the related findings in the Conclusions section.
Comment: The authors have used more than one reference style—none of which is the preferred MDPI style. Please redo the references to use the MDPI style.
Reply: We thank the reviewer for their comment regarding the reference style. We would like to clarify that we initially used EndNote configured with the MDPI style, and all references were generated automatically through the software rather than being entered manually. Any inconsistencies were unintentional and have been thoroughly reviewed. We have re-checked the reference list to ensure strict adherence to the MDPI style guidelines.
Round 2
Reviewer 1 Report
Comments and Suggestions for Authors
thank you!!! it is perfect!
Author Response
Comment: thank you!!! it is perfect!
Reply: Thank you very much for your positive feedback. Your encouraging comments affirm the quality and relevance of our work, and we greatly appreciate the time you took to review it.
Reviewer 2 Report
Comments and Suggestions for Authors
Thank you to the authors for the changes they made to their manuscript. All have improved it. A few remain
Line by line suggested edits
33-39 The authors define bullying. The definition does not clarify that the victim has less perceived power, the bully more, and the victim-bully less perceived power while identifying as a victim and more perceived power when engaging in actions recognized by a victim as bullying. Another class of individuals who sometimes engage in bullying are witnesses: those who report the bullying behavior and those who are complicit with it. Those who report such behavior do so intending to balance the perceived power dynamic of the bully and victim. Those who are complicit can feel powerless to change the dynamic—and fear that they may become victims if they intervene—or they may approve of the bullying behavior. Please investigate the definition of bullying more thoroughly, find a citation particular to bullying, and use the current citation regarding cyberbullying. The authors define overt and cyberbullying as the same. However, there are differences. There must be mention of these differences, providing citations using current research.
54 The authors must include a paragraph on why bullies bully others given that—in the paragraphs that follow—the research shows how harmful bullying is to the victim. Please provide recent research regarding the positive benefits of bullying for the bully.
90-100 and 102-106 These lines make several claims—all require citations to current research. Please provide appropriate citations for each sentence.
137-139 Please comment on those studies that address the unique socio-cultural context of Greece. In the Google Scholar search provided in the previous review of research on the topic since 2021, there were “About 10,100 results". Please choose the most relevant and explain how they differ from this study.
Author Response
Comment: 33-39 The authors define bullying. The definition does not clarify that the victim has less perceived power, the bully more, and the victim-bully less perceived power while identifying as a victim and more perceived power when engaging in actions recognized by a victim as bullying. Another class of individuals who sometimes engage in bullying are witnesses: those who report the bullying behavior and those who are complicit with it. Those who report such behavior do so intending to balance the perceived power dynamic of the bully and victim. Those who are complicit can feel powerless to change the dynamic—and fear that they may become victims if they intervene—or they may approve of the bullying behavior. Please investigate the definition of bullying more thoroughly, find a citation particular to bullying, and use the current citation regarding cyberbullying. The authors define overt and cyberbullying as the same. However, there are differences. There must be mention of these differences, providing citations using current research.
Reply: Thank you for your insightful comments regarding our definition of bullying. In response, we have revised the manuscript to provide a more comprehensive definition. We now explicitly state that bullying involves a power imbalance where the victim is perceived to have less power and the bully more. We also clarify the dynamics in victim-bully situations, where individuals may display power in some contexts while still identifying as victims. We have included a discussion on the role of bystanders, differentiating between those who intervene (with the intent to balance power) and those who are complicit, either due to feeling powerless or by approving the behavior. We have revised the text to highlight that while traditional (overt) bullying typically occurs in face-to-face settings and involves direct power imbalances, cyberbullying operates in digital spaces with distinct characteristics such as anonymity, rapid dissemination, and a modified power dynamic. To support these revisions, we have incorporated current literature that clarifies these aspects. The revised definition has been inserted in the first paragraph of the Introduction section (in red). We also added a sentence in the Limitations section (in red) to explicitly acknowledge the variability and subjectivity in defining bullying. We believe these changes address your concerns and strengthen the conceptual framework of our study.
Comment: 54 The authors must include a paragraph on why bullies bully others given that—in the paragraphs that follow—the research shows how harmful bullying is to the victim. Please provide recent research regarding the positive benefits of bullying for the bully.
Reply: Thank you for highlighting the need to address the potential benefits that bullies may perceive from their actions. In response, we have added a new paragraph (#4, in red) in the Introduction. This paragraph explains that, while bullying is harmful for victims, some bullies may derive perceived social benefits—such as enhanced status, popularity, and a sense of power—which may reinforce their behavior. We have supported these claims with recent research. We believe this addition provides a more balanced discussion of the dynamics underlying bullying behavior and appropriately addresses your comment.
Comment: 90-100 and 102-106 These lines make several claims—all require citations to current research. Please provide appropriate citations for each sentence.
Reply: Thank you for your valuable feedback. In response, we have revised the manuscript to include appropriate, current citations for each sentence in these sections. We believe that these modifications strengthen the manuscript and fully address your concerns.
Comment: 137-139 Please comment on those studies that address the unique socio-cultural context of Greece. In the Google Scholar search provided in the previous review of research on the topic since 2021, there were “About 10,100 results". Please choose the most relevant and explain how they differ from this study.
Reply: Thank you for your comment regarding the need to address studies that explore the unique socio-cultural context of Greece. In response, we have added (in red) the findings by Magklara et al. (2022) on the psychological impact of the spring 2020 lockdown on Greek children, as well as Morres et al. (2021), which examined changes in well-being in Greek adolescents. We explain that while these studies provide valuable insights into general mental health and lifestyle behaviors during the pandemic, our study extends the literature by incorporating a time-trend design that compares pre- and post-pandemic data on both traditional and cyberbullying experiences alongside emotional and behavioral difficulties. We believe this addition clearly delineates how our work builds upon and differs from previous studies within the Greek context.